## RESEARCH REPORT

# Fer3 is uniquely expressed in Notch$^{OFF}$ hemilineages, where it promotes interneuron identity

Kasey C. Drake, Sen-Lin Lai and Chris Q. Doe*

## ABSTRACT

In the *Drosophila* nervous system, neuroblasts (NBs) divide to produce a clone of neurons that establish distinct fates via precise timing and patterning of transcription factors (TFs). The final step in neurogenesis is a Notch/Numb asymmetric division that produces one daughter neuron that has active Notch signaling (Notch$^{ON}$) and one that does not (Notch$^{OFF}$). Notch$^{ON}$ neurons are well characterized, but the Notch$^{OFF}$ progeny are understudied due to lack of molecular markers. Here, we have identified Fer3 (forty-eight related 3) as a Notch$^{OFF}$-specific transcription factor expressed in the NB7-1, NB6-1 and NB5-2 lineages. Fer3 is inhibited by Notch signaling in post-mitotic neurons, thereby restricting its expression to Notch$^{OFF}$ neurons. In some contexts, Fer3 is a transcriptional repressor, but we find that Fer3 misexpression generates ectopic Dbx$^+$ neurons, and this is more penetrant when we misexpress a Fer3:activation domain fusion protein. Moreover, Fer3 is sufficient to induce Dbx expression in a neuroblast lineage that does not endogenously express Dbx or Fer3. This work presents the first known Notch$^{OFF}$ exclusive TF in the developing embryonic *Drosophila* ventral nerve cord.

KEY WORDS: Notch, Hemilineage, Fer3, *Drosophila*, Neurogenesis, Neuroblast

## INTRODUCTION

Neural diversity is necessary for the formation and function of complex nervous systems across animal species, from *Drosophila* to mammals. In the *Drosophila* ventral nerve cord (VNC), neural diversity is achieved by the integration of three developmental steps. The first two steps establish spatial and temporal identity, which describe the inheritance of distinct transcription factor (TF) profiles in each intermediate progenitor (ganglion mother cell; GMC) based on the location and time of their division from the neuroblast (NB) (Fig. 1A) (Doe, 2017; Skeath and Thor, 2003). The combination of spatial and temporal patterning help establish unique identities in neurons, generating 30 distinct NBs per hemisegment (Broadus et al., 1995). The third step of neurogenesis is the division of the GMC to generate two distinct neurons: one with active Notch signaling (Notch$^{ON}$) and one lacking Notch signaling (Notch$^{OFF}$) (Skeath and Doe, 1998; Spana et al., 1995). This terminal division is asymmetric, with the Notch inhibitor Numb segregating to one side of the GMC

Institute of Neuroscience, Howard Hughes Medical Institute, University of Oregon, Eugene, OR 97403, USA.

*Author for correspondence (cdoe@uoregon.edu)

S.-L.L., 0000-0002-7531-283X; C.Q.D., 0000-0001-5980-8029

such that it is inherited by the Notch$^{OFF}$ progeny. In this way, each NB lineage generates a Notch$^{ON}$ hemilineage and a Notch$^{OFF}$ hemilineage.

Here, we focus on the third step: Notch$^{ON}$/Notch$^{OFF}$ GMC asymmetric division; earlier roles for Notch in NB segregation and type 1>0 NB division switch are addressed elsewhere (Skeath and Thor, 2003; Baumgardt et al., 2014). Previous work has shown that hemilineage identity is necessary for establishing molecular and morphological differences between sibling neurons that share a spatial and temporal identity (Harris et al., 2015; Mark et al., 2021; Truman et al., 2010). In the NB7-1 lineage, the U1-U5 motor neurons are all Notch$^{ON}$, and their Notch$^{OFF}$ siblings have interneuron fates (Skeath and Doe, 1998; Spana et al., 1995). Part of the reason we know relatively little about how sibling neurons acquire distinct cell fates is that, historically, analysis of motor neuron specification has been a major focus of research, and their interneuron siblings have few known molecular markers. Additionally, the lack of known genes specifically expressed in Notch$^{OFF}$ neurons makes them challenging to access genetically. To our knowledge, there are currently no known Notch$^{OFF}$-specific genes and crucially no pan-Notch$^{OFF}$-specific genes; the closest is *Dbx*, which is expressed in many Notch$^{OFF}$ interneurons, but also in the Notch$^{ON}$ RP2 sib interneuron (Lacin et al., 2009; Skeath and Doe, 1998; Spana et al., 1995). This poses a challenge for studying the mechanisms underlying hemilineage specification.

Here, we have characterized the embryonic VNC expression and function of the first known gene expressed solely in embryonic Notch$^{OFF}$ neurons, the *Drosophila* Forty-eight related 3 (Fer3) TF. We show that Fer3 is expressed in a subset of neurons in the NB7-1, NB5-2 and NB6-1 lineages. Additionally, we show that Fer3 is inhibited by Notch signaling, making its expression exclusive to Notch$^{OFF}$ hemilineages. Finally, we discovered that Fer3 functions as an activator of Dbx and is sufficient to induce Dbx expression in ectopic contexts.

## RESULTS AND DISCUSSION

### Fer3 is a transcription factor expressed in the NB7-1, NB5-2 and NB6-1 lineages

Fer3 is a basic helix-loop-helix (bHLH) TF that was recently identified as a top enriched gene in the NB7-1 lineage (Seroka et al., 2022). To determine the protein expression profile of Fer3 in the *Drosophila* CNS, we generated a Fer3-specific antibody. We assayed Fer3 expression in NBs, GMCs and neurons. We found that Fer3 was expressed in Elav$^+$ neurons but not in Ase$^+$ NBs or GMCs (Fig. 1B-C″). Fer3 showed expression in a posterior-medial neuron cluster in each hemisegment at embryonic stage 11. By stage 17, Fer3 was expressed in an average of 26±2 cells per hemisegment (*n*=88; 6 embryos). We conclude that Fer3 expression is limited to post-mitotic neurons.

The Fer3 expression pattern was consistent in both thoracic and abdominal segments, and persisted through first instar larvae. Since the expression patterns were similar, we chose to analyze

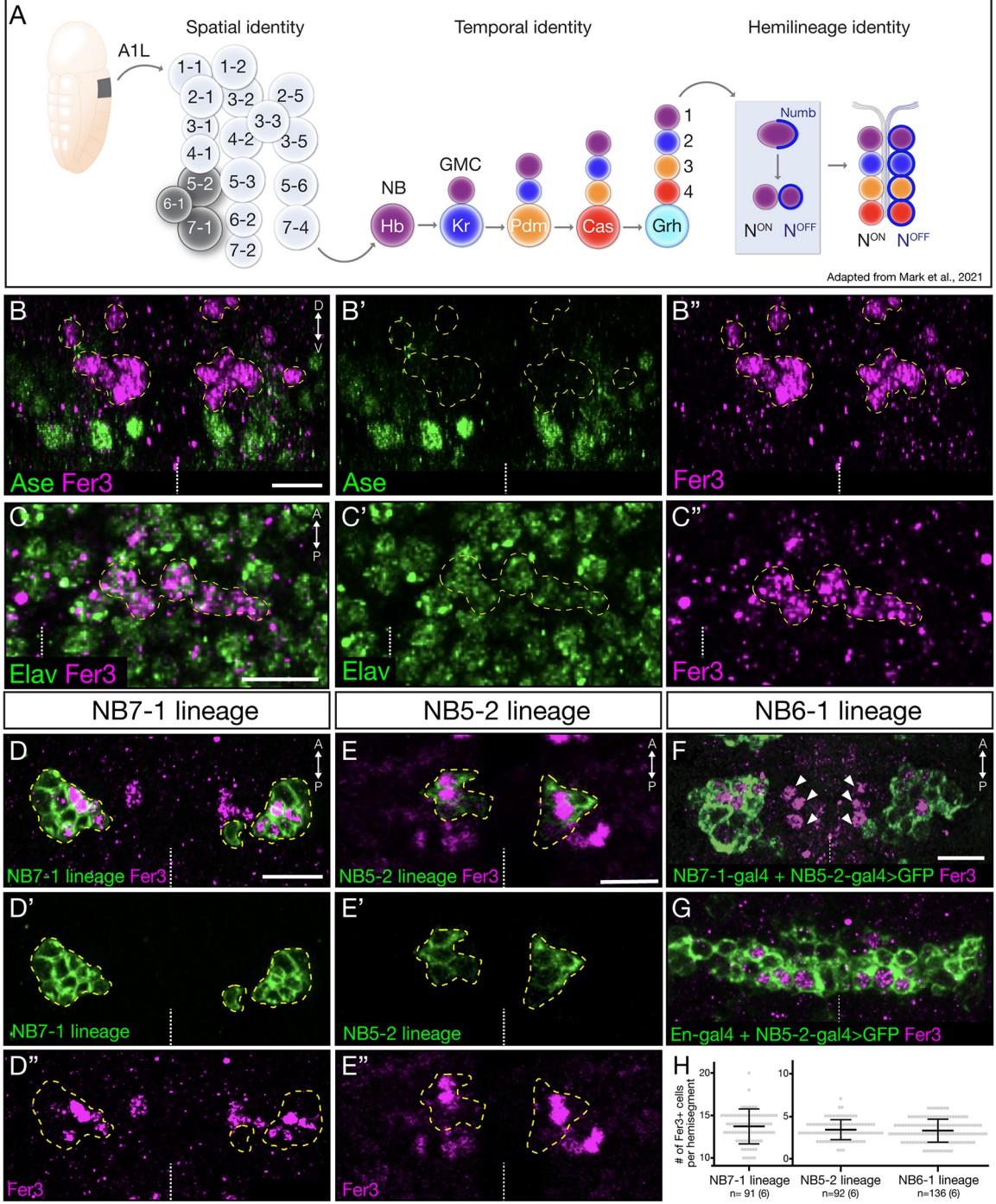

**Fig. 1. Fer3 is a transcription factor expressed in a subset of the NB7-1 and NB5-2 progeny.** (A) Schematic of *Drosophila* embryonic VNC neural diversity generated by spatial, temporal and hemilineage identities. (B-G) All images show a dorsal view of stage 14 (B,C) or stage 17 (D-G) *Drosophila* embryo abdominal segments. White dashed line indicates midline. Scale bars: 10 µm. (B-B″) Expression of Fer3 and Asense. Posterior view; stage 14 embryo; yellow dashed lines outline Fer3$^+$ cells. (C-C″) Expression of Fer3 and Elav. Stage 14 embryo. Yellow dashed line outlines Fer3$^+$ cells. (D-D″) Expression of Fer3 in the NB7-1 lineage. Genotype: *NB7-1-Gal4,UAS-myr::smGdP-HA*. Yellow dashed lines outline the NB7-1 lineage. (E-E″) Expression of Fer3 in the NB5-2 lineage. Genotype: *NB5-2-Gal4,UAS-myr::smGdP-HA*. Yellow dashed lines outline the NB5-2 lineage. (F) Expression of Fer3 outside the NB7-1 and NB5-2 lineages. There is a small cluster of Fer3$^+$ cells (arrowheads) outside both lineages. Genotype: *NB5-2-Gal4,NB7-1-Gal4,UAS-myr:: smGdP-HA*. (G) Expression of Fer3 in the VNC. All Fer3$^+$ cells are in NB5-2 and En-Gal4$^+$ lineages. Genotype: *NB5-2-Gal4,en-Gal4,UAS-myr::smGdP-HA*. (H) Quantification of Fer3$^+$ cells. Each dot indicates one hemisegment, all A1-A6. The thick bar indicates the average; error bars indicate the s.d. *n*, number of hemisegments. The numbers of embryos are in parenthesis.

abdominal segments A1-A6. We hypothesized that the cluster of Fer3$^+$ post-mitotic neurons are derived solely from the NB7-1 lineage. Using a NB7-1-lineage specific GAL4 (Anderson et al., 2025) driving a *UAS-HA*, we found an average of 14±2 Fer3$^+$ cells

in each NB7-1 lineage (Fig. 1D-D″,H). Additionally, we observed two clusters of Fer3$^+$ neurons located outside the NB7-1 lineage. One cluster was located anterior to the NB7-1 lineage; using a NB5-2-specific Gal4 driver (Pollington and Doe, 2025) to drive *UAS-HA*

DEVELOPMENT

generated an average of 3±1 NB5-2-derived Fer3$^+$ cells per hemisegment (*n*=92; Fig. 1E-E″,H). The final cluster was also located close to the NB7-1 lineage. When we used both NB7-1 and NB5-2 Gal4 drivers to drive *UAS-HA*, we detected 4±2 Fer3$^+$ cells that did not express either NB7-1 or NB5-2 transgene (Fig. 1F,H). To further identify what spatial factors defined this third cluster, we used NB5-2 and *engrailed-gal4* drivers to drive the expression of *UAS-HA*, resulting in all Fer3$^+$ cells being labeled (Fig. 1G). The final cluster resided in the most anterior-medial region of Engrailed$^+$ cells, leading us to infer that the third cluster of Fer3$^+$ cells is from NB6-1. In support, Fer3 was previously shown to be in the NB6-1 lineage (Soffers et al., 2025). We conclude that Fer3 is expressed in 14 cells in the NB7-1 lineage, and in about four cells in each of the NB5-2 and NB6-1 lineages.

Fer3 is expressed in only the NB7-1, NB6-1, and NB5-2 lineages. Why only these three NBs? All three express the same two spatial patterning genes: *ventral nervous system defective* (*vnd*) and *gooseberry* (*gsb*) (Broadus et al., 1995; McDonald et al., 1998; Skeath et al., 1994, 1995); we propose that their shared spatial TFs are permissive for Fer3 expression. There are no other obvious features uniquely shared by these three lineages: all produce Notch$^{ON}$/Notch$^{OFF}$ siblings; NB7-1 produces five Eve$^+$ Unc-4$^+$ Notch$^{ON}$ U1-U5 motor neurons and their Notch$^{OFF}$ Dbx$^+$ sibling interneurons (Lacin et al., 2009); and NB5-2 produces primarily

Notch$^{ON}$/Notch$^{OFF}$ interneurons (Mark et al., 2021). NB6-1 produces ~26 interneurons (Bossing et al., 1996; Schmid et al., 1999), some of which are Dbx$^+$ (Lacin et al., 2009).

## Fer3 is expressed in both early- and late-born neurons

We focus our subsequent analysis on the well-characterized NB5-2 and NB7-1 lineages because we cannot assay NB6-1 due to lack of genetic tools. NB7-1 and NB5-2 both undergo the canonical temporal TF (TTF) cascade, in which they express Hunchback, Krüppel, Pdm and Castor (Isshiki et al., 2001). To determine whether Fer3$^+$ neurons derived from one or more temporal windows, we combined TTF antibody staining with our NB7-1-Gal4 and NB5-2-Gal4 lines. We found that, in the NB7-1 lineage, most hemisegments (76%) contained at least one Hunchback$^+$/Fer3$^+$ cell, and nearly all had at least one Krüppel$^+$, Pdm$^+$ or Castor$^+$ neuron (>83%) (Fig. 2A-E). In the NB5-2 lineage, 91% of analyzed hemisegments had at least one Hunchback$^+$/Fer3$^+$ cell, 78% had at least one Krüppel$^+$/Fer3$^+$ cell, 88% had at least one Pdm$^+$/Fer3$^+$ cell, and 72% had at least one Castor$^+$/Fer3$^+$ cell (Fig. 2F-J). We conclude that in the NB5-2 and NB7-1 lineages, Fer3 is expressed in ~1 neuron in each TTF window.

## Fer3 is specifically expressed in Notch$^{OFF}$ neurons

The final stage of neural diversity generation consists of an asymmetric division of the Notch inhibitor Numb, resulting in two

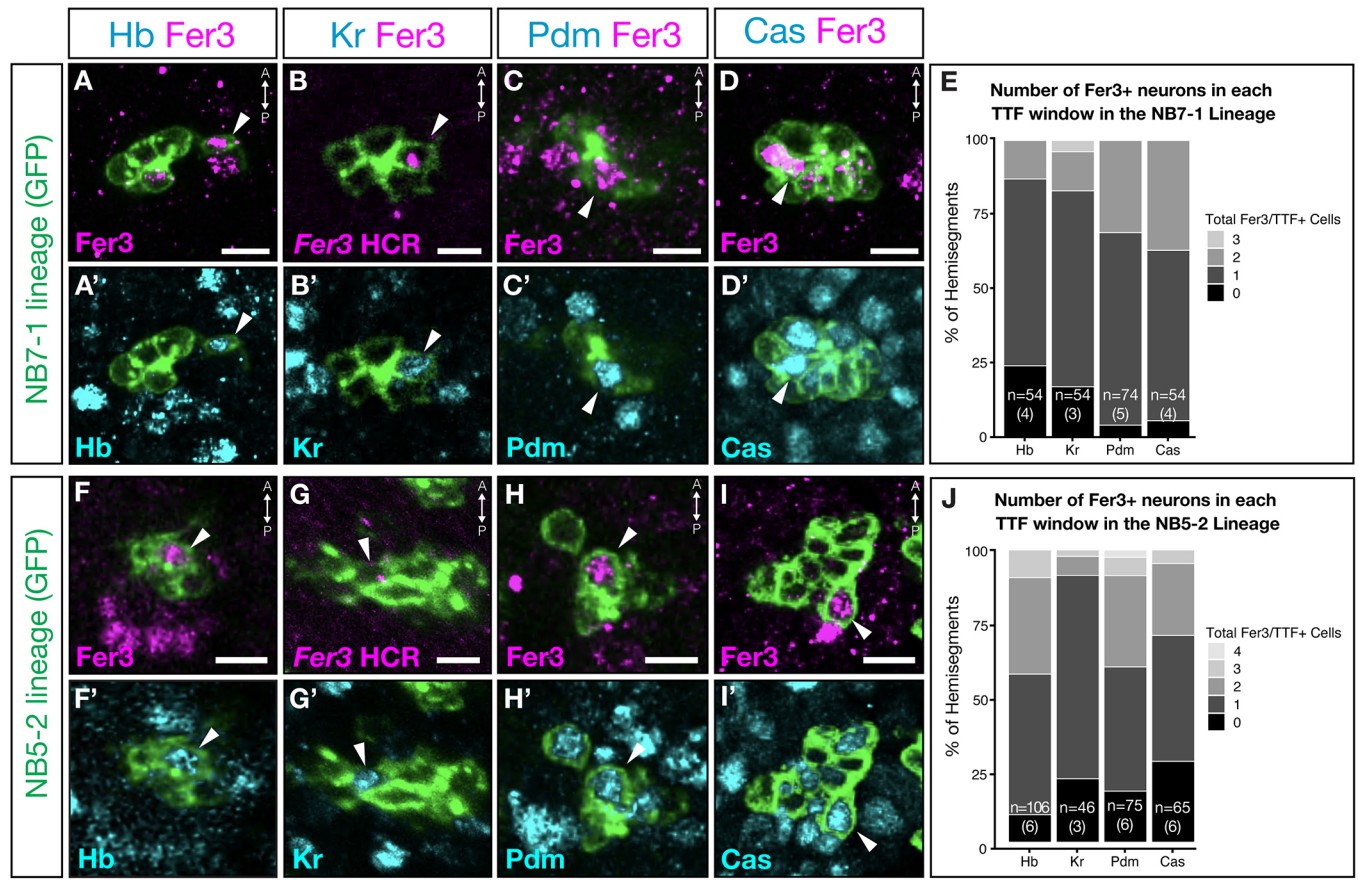

**Fig. 2. Fer3 is expressed in both early- and late-born *Drosophila* neurons.** All images show a dorsal view of an optical section from one abdominal hemisegment, A1-6; stage 16 embryos. Lateral is towards the left; midline is towards the right. Since Krüppel and Fer3 antibodies were both made in the same species, *Fer3* HCR (hybridization chain reaction; a method for visualizing RNA in cells) was used to identify colocalization of Krüppel and Fer3 expression. (A-D′,F-I′) Fer3 and temporal transcription factor expression in NB7-1 and NB5-2 lineages. NB7-1 genotype: *NB7-1-Gal4,UAS-myr::smGdP-HA*. NB5-2 genotype: *NB5-2-Gal4,UAS-myr::smGdP-HA*. Scale bars: 5 µm in A,A′,C-D′,F,F′,H-I′; 2.5 µm in B,B′,G,G′. (E,J) Quantification of cells co-expressing Fer3 and temporal transcription factors in the NB7-1 (E) and NB5-2 (J) lineages. *n*, number of hemisegments. The numbers of embryos are in parenthesis.

daughter cells where one is Notch[ON] and one is Notch[OFF] (Fig. 1A) (Skeath and Doe, 1998; Spana et al., 1995). To determine if Fer3 is expressed in the Notch[ON] or Notch[OFF] population, we assayed the

Notch[ON] downstream factor Hey (Monastirioti et al., 2010). In both the NB7-1 and NB5-2 lineages, Fer3 expression was mutually exclusive with Hey expression (Fig. 3A-B). The third cluster of

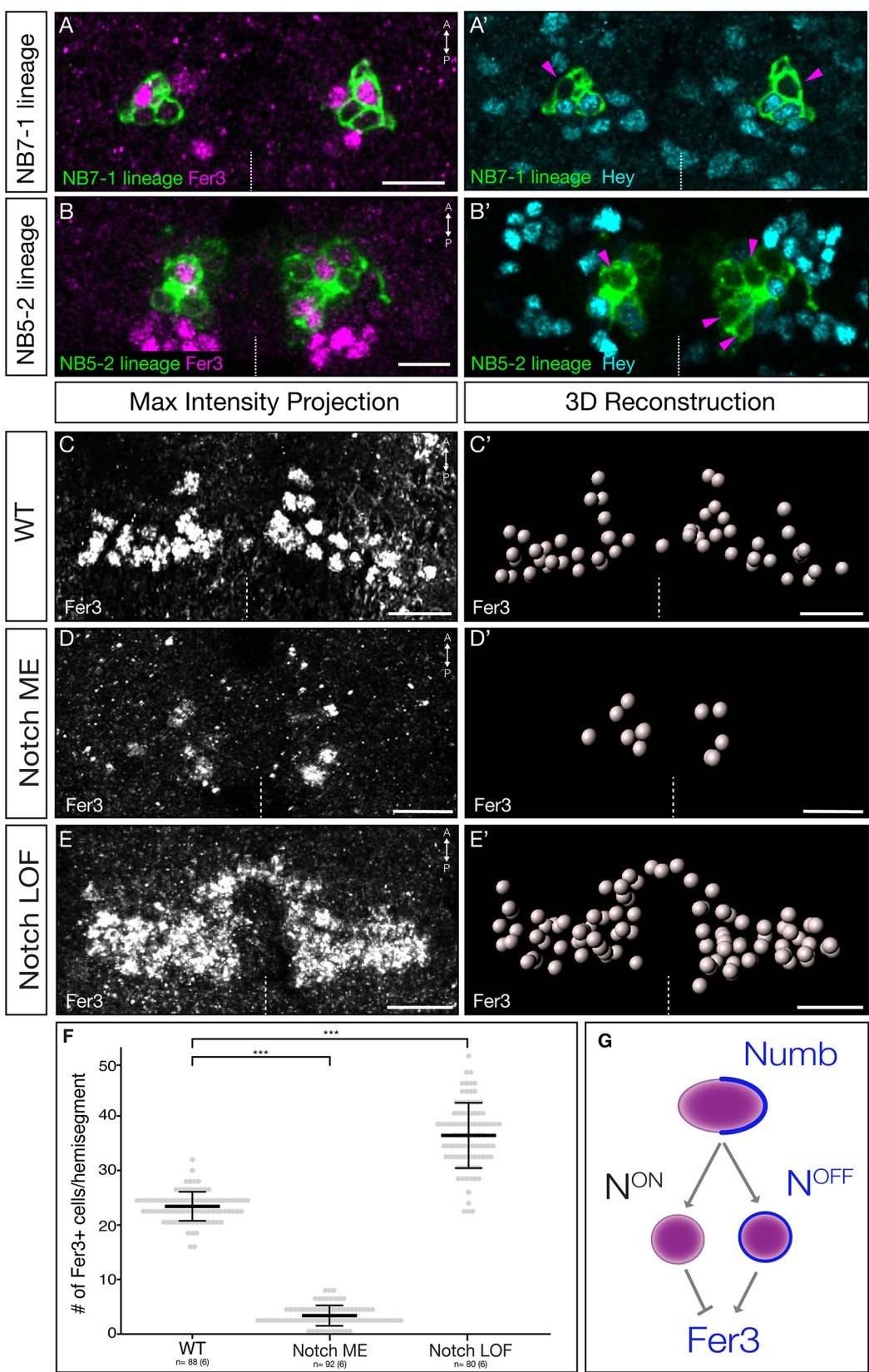

Fig. 3. See next page for legend.

**Fig. 3. Fer3 is only expressed in Notch$^{OFF}$ neurons.** (A-E′) All images are dorsal views of a single abdominal segment in a stage 17 *Drosophila* embryo. Dashed line indicates the midline. Scale bars: 10 µm. (A,A′) One optical section of Fer3 (A) and Hey (A′) expression in the NB7-1 lineage. Magenta arrowheads (A′) indicate Fer3$^+$ cells. Genotype: *NB7-1-Gal4,UAS-myr::smGdP-HA*. (B,B′) One optical section of Fer3 (B) and Hey (B′) expression in the NB5-2 lineage. Magenta arrowheads (B′) indicate Fer3$^+$ cells. Genotype: *NB5-2-Gal4,UAS-myr::smGdP-HA*. (C-E′) Expression of Fer3 in wild type, Notch activity misexpression and Notch loss-of-function embryos. (C) Maximum intensity projection of Fer3 expression. Genotype: *yw*. (C′) Imaris spot 3D reconstruction of the segment shown in C. (D) Maximum intensity projection of Fer3 staining in Notch activity misexpression (ME). Genotype: *elav-Gal4,UAS-N$^{intra}$*. (D′) Imaris spot 3D reconstruction of the segment shown in D. Note the loss of Fer3$^+$ cells. (E) Maximum intensity projection of Fer3 expression in Notch loss-of-function. Genotype: *spdo* mutant. (E′) Imaris spot 3D reconstruction of the segment shown in E. Note the increase in Fer3$^+$ cells. (F) Quantification of Fer3$^+$ cells per hemisegment in wild type, Notch ME and Notch loss of function (*sanpodo* mutant). Each dot represents one hemisegment, all A1-A6. The thick bar indicates the average; error bars indicate the s.d. Wild type, 23.4±2.7 cells; Notch ME, 3.4±1.9 cells; Notch loss of function, 36.4 ±6.0 cells. ns, not significant ($P$>0.05). ***$P$<0.001. (G) Schematic model of the asymmetric Notch/Numb division with Fer3. *n*, number of hemisegments. The numbers of embryos are in parenthesis.

Fer3$^+$ cells (NB6-1) also showed mutual exclusivity between Fer3 and Hey expression (data not shown). From this, we conclude that Fer3 is specifically expressed in Notch$^{OFF}$ neurons.

### Reduced Notch levels increase the number of Fer3$^+$ neurons, whereas increased Notch levels reduce the number of Fer3$^+$ neurons

Since Fer3 is specifically expressed in Notch$^{OFF}$ cells, we hypothesized that reduced Notch activity would increase the number of Fer3$^+$ cells, whereas increased Notch activity would reduce the number of Fer3$^+$ cells. To increase Notch activity, we misexpressed Notch$^{intra}$ – a constitutively active Notch transgene (Struhl and Adachi, 1998) – in all cells and quantified the total number of Fer3 cells per hemisegment. In controls, we saw the expected ~23 Fer3$^+$ cells per hemisegment (Fig. 3C,C′). In contrast, Notch$^{intra}$ misexpression led to a significant loss of Fer3$^+$ cells to 3 cells per hemisegment (Fig. 3D,D′,F). Next, we assayed *sanpodo* mutants, which are known to block Notch signaling in sibling cells (Skeath and Doe, 1998; Spana et al., 1995), and found a significant increase of Fer3$^+$ cells to ~36 per hemisegment (Fig. 3E-E′,F). These results demonstrate that expression of Fer3 is influenced by the Notch/Numb asymmetric division and explain how Fer3 cells all have the Notch$^{OFF}$ fate (Fig. 3G).

### Fer3 activates Dbx in the Notch$^{OFF}$ NB7-1 and 5-2 hemilineages

We next wanted to explore the functional role of Fer3. Because Fer3 is directly regulated by Notch (see above), we hypothesized that Fer3 may regulate Dbx, a known Notch$^{OFF}$ expressed transcription factor (Lacin et al., 2009). We therefore assayed Dbx$^+$ cells in Fer3 mutant and misexpression embryos. To generate a Fer3 loss-of-function allele, we inserted an FRT.stop cassette in front of the endogenous Fer3 locus to block Fer3 transcription. In controls, we saw the previously reported ~8 Dbx cells per hemisegment in the NB7-1 lineage and ~2 Dbx cells in the NB5-2 lineage (Lacin et al., 2009) (Fig. 4A,A′,F,F′). In contrast, Fer3 loss of function embryos showed reduced numbers of Dbx$^+$ cells in each lineage, from ~8 to ~3 Dbx$^+$ cells per hemisegment in the NB7-1 lineage (Fig. 4B-B′,E) and from ~2 to 0 Dbx$^+$ cells per hemisegment in the

NB5-2 lineage (Fig. 4G,G′,J). The total number of neurons within each hemisegment was the same as controls, ruling out cell death (Fig. 4E,J). Additionally, the loss of Fer3 expression did not induce ectopic Eve expression in sibling Notch$^{OFF}$ neurons in the NB7-1 lineage, indicating the fate of these interneurons was not transformed into the Notch$^{ON}$ Eve$^+$ U1-U5 motor neurons, and supporting our finding that Fer3 operates downstream of Notch (Fig. 3). Next, we misexpressed Fer3 in the NB7-1 or NB5-2 lineages. The misexpression of Fer3 led to a slight but significant increase in total Dbx$^+$ cells, increasing the totals to from ~8 to ~10 Dbx$^+$ cells per hemisegment in the NB7-1 lineage (Fig. 4C,C′,E) and from ~2 to ~6 cells per hemisegment in the NB5-2 lineage (Fig. 4H,H′,J). The similar numbers indicate that cell death and over-proliferation were not relevant (Fig. 4E,J).

The observation that Fer3 activates Dbx expression was initially surprising, because previous studies of Fer3 showed that it acts as a transcriptional repressor (Verzi et al., 2002). To confirm Fer3 was acting as an activator of Dbx rather than repressing a repressor of Dbx, we created a Fer3 protein containing a p65 activation domain, which should turn Fer3 into a constitutive transcriptional activator (subsequently, Fer3:AD). If Fer3 was repressing a repressor of Dbx, we would expect the Fer3:AD protein to increase repressor expression and thus decrease Dbx expression, whereas if Fer3 is acting as an activator we would expect the Fer3:AD protein to increase Dbx expression. Indeed, we observed a dramatic increase in Dbx$^+$ cells in both lineages when Fer3:AD was expressed, from ~8 to ~18 Dbx$^+$ cells per hemisegment in the NB7-1 lineage (Fig. 4D-D′,E) and from ~3 to ~12 in the NB5-2 lineage (Fig. 4I-I′,J). There was no significant change in the total number of cells in both the NB7-1 and NB5-2 lineages during Fer3: AD misexpression, showing that Fer3 induced a change of fate (Fig. 4E,J). Based on these results, we conclude that Fer3 functions as a transcriptional activator promoting expression of Dbx in the NB7-1 and NB5-2 lineages.

Previous studies have identified Fer3 as a transcriptional repressor (Verzi et al., 2002), whereas we show it is activator, indicating that its activity is context dependent, a characteristic shared by other TFs like Hunchback (Hulskamp et al., 1990; Tran et al., 2010). Since Fer3 is a bHLH protein, it is likely that Fer3 dimerizes with co-regulators in different contexts to induce distinct functions. This hypothesis explains why some, but not all, Fer3$^+$ cells are Dbx$^+$, and why adding an ectopic AD, essentially by-passing any co-regulators, had such a dramatic effect.

### Fer3 is sufficient to induce ectopic Dbx expression in the NB5-6 lineage

The NB6-2 and NB4-2 lineages also produce a subset of Dbx$^+$ cells (Lacin et al., 2009), but do not express Fer3 (Fig. 1). Because Fer3 is sufficient to induce Dbx expression in both the NB7-1 lineage and NB5-2 lineages but is not necessary in the NB6-2 or NB4-2 lineages, we hypothesize that Fer3 activation of Dbx is context dependent. To test this, we misexpressed Fer3 and the Fer3:AD in the NB5-6 lineage (Baumgardt et al., 2009), which in wild type does not express Fer3 or Dbx. In controls, we saw the predicted 0 Dbx cells in all NB5-6 hemisegments (Fig. 4K-K″,N). When Fer3 was expressed, we saw ~3 Dbx$^+$ cells per hemisegment in the NB5-6 lineage (Fig. 4 L-L″,N) and when Fer3:AD was expressed, we saw ~16 Dbx$^+$ cells per hemisegment in the NB5-6 lineage (Fig. 4M-M″,N). Total cell counts in the Fer3 misexpression flies were not significantly different from wild type, suggesting the misexpression phenotype was not a result of ectopic cells (Fig. 4N). Conversely, the Fer3:AD misexpression flies had a significant

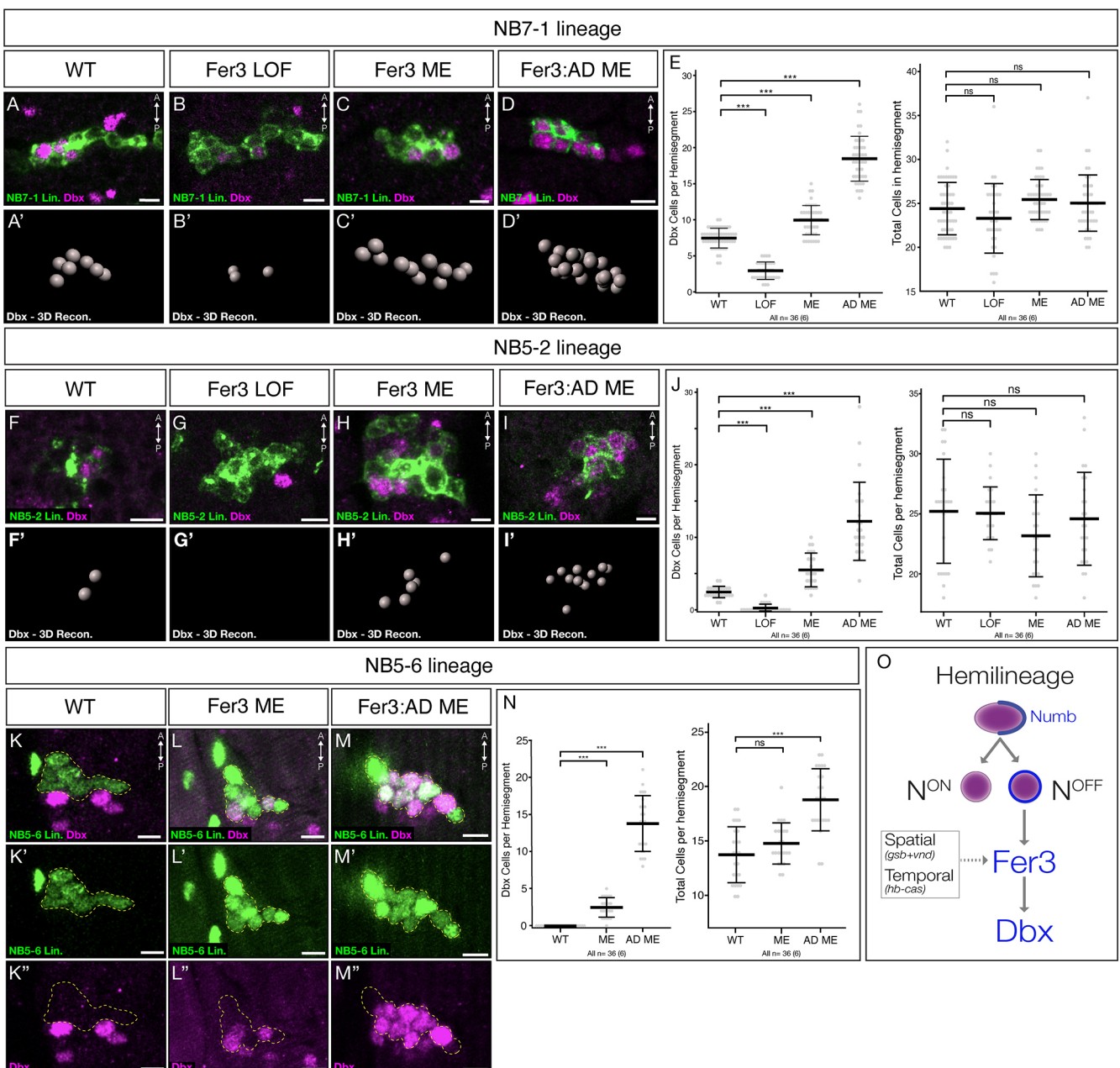

**Fig. 4. Fer3 activates expression of Dbx.** (A-D′,F-I′,K-M″) All images are a dorsal view of a 1 µm thick optical section of a single hemisegment in a stage 17 *Drosophila* embryos. Midline is towards the right; lateral is towards the left. Scale bars: 5 µm. (E,J,N) For all quantifications, each dot represents one hemisegment, all A1-A6 unless otherwise noted. The thick bar indicates the average; error bars indicate the s.d. ns, not significant (*P*>0.05). ****P*<0.001. (A-D) Expression of Dbx in NB7-1 lineage in wild-type, Fer3 loss of function and Fer3 misexpression embryos. (A′-D′) Imaris spot 3D reconstructions of all Dbx+ cells from the hemisegment in A-D. Genotypes: wild type, *NB7-1-Gal4^{KZ},UAS-myr::smGdP-HA*; Fer3 loss of function (Fer3 LOF), *NB7-1-Gal4^{KZ},UAS-myr::smGdP-HA, Fer3 mutant*; Fer3 misexpression (Fer3 ME), *NB7-1-Gal4^{KZ},UAS-myr::smGdP-HA,UAS-Fer3*; Fer3-activator misexpression (Fer3:AD ME), *NB7-1-Gal4^{KZ},UAS-myr::smGdP-HA,UAS-Fer3:AD*. (E) Quantification. Dbx+ cells in NB7-1 lineage: wild type, 7.5±1.4; Fer3 loss of function, 3.9±1.7; Fer3 ME, 10.0±2.0; Fer3:AD ME, 18.5±3.1. Total cells in NB7-1 lineage: wild type, 24.4±3.0; Fer3 loss of function, 25.1±5.5; Fer3 ME, 25.4±2.3; Fer3:AD ME, 25.0±3.2. (F-I) Expression of Fer3 in NB5-2 lineage in wild type, Fer3 loss of function and Fer3 misexpression embryos. (F′-I′) Imaris spot 3D reconstructions of all Dbx+ cells from the hemisegment in F-I. Genotypes: wild type, *NB5-2-Gal4,UAS-myr::smGdP-HA*; Fer3 loss of function, *NB5-2-Gal4,UAS-myr::smGdP-HA, Fer3 mutant*; Fer3 ME, *NB5-2-Gal4,UAS-myr::smGdP-HA,UAS-Fer3*; Fer3:AD ME, *NB5-2-Gal4,UAS-myr::smGdP-HA,UAS-Fer3:AD*. (J) Quantification. Dbx+ cells in NB5-2 lineage: wild type, 0±0; Fer3 ME, 2.5±1.3; Fer3:AD ME, 15.6±3.2. Total cells in NB5-2 lineage: wild type, 13.8±2.6; Fer3 ME, 14.9±1.9; Fer3:AD ME, 19.0±3.6. (K-M) Expression of Dbx in NB5-6 lineage in thoracic segments of wild type (K-K″), Fer3 misexpression (L-L″) and Fer3:AS misexpression (M-M″) embryos. Genotypes: wild type, *NB5-6-Gal4,UAS-RedStinger*; Fer3 ME, *NB5-6-Gal4,UAS-RedStinger,UAS-Fer3*; Fer3:AD ME, *NB5-6-Gal4,UAS-RedStinger,UAS-Fer3:AD*. (N) Quantification. Dbx+ cells in T1-3 of NB5-6 lineage: wild type, 7.5±1.4; Fer3 ME, 10.0±2.0; Fer3:AD ME, 18.5±3.1. Total cells in T1-3 of NB5-6 lineage: wild type, 24.4±3.0; Fer3 ME, 25.4±2.3; Fer3:AD ME, 25.0±3.2. (O) Summary of findings and hypotheses. Solid arrows indicate proven regulation; dashed arrow indicates hypothesized regulation. *n*, number of hemisegments. The numbers of embryos are in parenthesis.

increase in total cell number, indicating the Fer3:AD misexpression is not only transforming neuronal identity, but may also increase the length of the lineage or extend the expression of the NB5-6-Gal4 (Fig. 4N). This expansion of Dbx$^+$ cells in the NB5-6 lineage shows that Fer3 is sufficient to induce Dbx expression in neuroblasts that normally do not have Dbx$^+$ progeny.

## Conclusions

These findings led to our final model (Fig. 4O) describing Fer3 expression: (1) since NB7-1, NB5-2 and NB6-1 share the spatial factors Gsb and Vnd, we hypothesize that these spatial factors are specifically permissive to Fer3; (2) similarly, we hypothesize that the Hunchback>Castor TTF cascade is permissive for Fer3 expression, with later TTFs not allowing Fer3 expression in some lineages, such as NB5-2; and (3) Notch signaling inhibits Fer3 to restrict its expression to Notch$^{OFF}$ hemilineages only, providing a specific context in which Fer3 is co-regulated by other proteins to further specify downstream targets such as Dbx. These results lead to further open questions: Are Gsb and Vnd both required for Fer3 expression? Are the Hunchback>Castor TTFs required for Fer3 expression? Does the bHLH Fer3 TF work with one or more co-factors? What are the Fer3 downstream targets, in addition to Dbx? Are there other TFs like Fer3 that are Notch$^{OFF}$-specific in other NB lineages? Our characterization of Fer3 opens the door for further work on the Notch$^{OFF}$ hemilineage by providing genetic access to a subset of embryonic Notch$^{OFF}$ neurons.

## MATERIALS AND METHODS

### Fly stocks

#### Animal preparation

*Drosophila melanogaster* was used in all experiments. Male and female flies were used for all experiments. Flies were kept and maintained at 25°C unless stated otherwise. Stocks used can be found in Table S1.

#### Embryo sample preparation

Embryos were collected overnight on 3.0% agar apple juice caps with yeast paste at 25°C. Embryonic stages were identified post-hoc by analyzing gut morphology: at stage 14, gut is tube shaped; at stage 15, gut is heart shaped; at stage 16, gut is coiled three times; at stage 17, gut is coiled four times.

Embryos were transferred from apple caps into collection baskets and rinsed with deionized H$_2$O. Embryos were dechorionated in 100% bleach (Clorox) for 4 min with gentle agitation. Dechorionated embryos were rinsed with deionized H$_2$O for 1 min. Embryos were fixed 20 mins in 2 ml Eppendorf tubes containing equal volumes of Heptane (Fisher Chemical, H3505K-4) and 4% PFA diluted in PEM. Fix was removed and 500 µl of methanol (Fisher Chemical, A412P-4) was added to each tube. Heptane was removed and 500 µl of methanol were then added. Tubes were then subject to vigorous agitation for 1 min in a step required for removing the vitelline membrane. Nearly all liquid was removed from the tubes, leaving the embryos. Embryos were rinsed in methanol three times before being stored at −20°C.

#### Fer3 transgenic flies and antibody

Fer3 mutant was generated by knocking in the (FRT-stop)-T2A-lexA.p65-T2A sequence [cloned into pHD-DsRed (RRID:Addgene plasmid #51434)] at the N terminus of Fer3 open reading frame (ORF). This was achieved using the gRNAs TGCATGACTGTCTATTGTAT and CATGTAGGTGGGCTGGTCAA that were cloned into pCFD4 (RRID: Addgene plasmid #49411). Injection of constructs into *yw;nos-Cas9*(II-attP40) and transformant selections were performed by BestGene. The UAS-Fer3 construct was generated by cloning 2xHA fused at the C terminus in frame with the Fer3 ORF into attB-pUAST. UAS-Fer3-p65 was created by fusing 2xFlag-p65 with the Fer3 ORF at the N terminus and cloning it into attB-pUAST. UAS-Fer3 and UAS-Fer3-p65 constructs were inserted into attP40. Injections and transformant selections were performed by

BestGene. The Fer3 antibody was raised against a synthesized peptide of the full Fer3 protein in guinea pig by GenScript (Piscataway, NJ, USA).

### Immunohistochemistry

Embryos stored in methanol were washed with 1×PBS/0.1% Triton X solution three times for 10 min each on a rocker at room temperature. After washes, embryos were rocked in a blocking solution (5% normal donkey serum in 1×PBS/0.1% Triton X) for 30 min before incubation in primary antibodies overnight at 4°C. After primary incubation, embryos were washed three times for 10 min in 1×PBS/0.1% Triton X solution and then incubated in secondary antibodies at a 1:200 concentration for 2 h at room temperature. After secondary antibody incubation, embryos were washed a final three times for 10 min each in 1×PBS/0.1% Triton X before equilibration in 50% glycerol/PBS and then 90% glycerol/PBS. Embryos were stored at 4°C for no longer than 1 week before imaging. Antibodies are listed in Table S1.

### Hybridization chain reaction

We followed the protocol described by Bruce et al. (2021) for hybridization chain reaction specifically in *Drosophila* embryos. The HCR probe and amplifier specifics are listed in Table S1.

### Confocal microscopy

Fixed preparations were imaged with a Zeiss LSM 900 laser scanning confocal equipped with an Axio Imager.Z2 microscope. A 10×/0.3 EC Plan-Neofluar M27 or a 40×/1.40 NA Oil Plan-Apochromat DIC M27 objective lens was used. The software program used was Zen 3.6 (blue edition; Zeiss).

### Image processing and analysis

Cell counting was carried out using Napari (Sofroniew et al., 2025) with the napari-3d-counter plugin (doi:10.5281/zenodo.17436864). Images in figures were prepared using Imaris 10.0.1. Scale bars are given for a single slice in all single slice images and from all stacks within maximum intensity projections images. Pixel brightness was adjusted in images for clearer visualization; all adjustments were made uniformly over the entire image, and uniformly across wild-type samples and corresponding control and experimental samples. Adobe Illustrator 2024 was used for figure formatting.

### Statistical analyses

Statistics were computed using R 4.4.1. All statistical tests used were one-way ANOVA with Tukey's post-hoc test. Plots were generated using R.

**Acknowledgements**
We thank Haluk Lacin, Megan Radler, Kristen Lee and Judith Eisen for constructive comments on the manuscript. We thank Kate O'Connor-Giles and Simon Bullock for plasmids, which were obtained from Addgene. Stocks obtained from the Bloomington Drosophila Stock Center (NIH P40OD018537) were used in this study. Antibodies obtained from the Developmental Studies Hybridoma Bank, created by the NICHD of the NIH and maintained at the University of Iowa, Department of Biology, Iowa City, IA were used in this study. FlyBase was used extensively in this work; FlyBase's data and annotations are available with a CC BY 4.0 license.

**Competing interests**
The authors declare no competing or financial interests.

**Author contributions**
Conceptualization: K.C.D., S.-L.L., C.Q.D.; Data curation: K.C.D., S.-L.L.; Formal analysis: K.C.D., S.-L.L.; Funding acquisition: C.Q.D.; Investigation: K.C.D., S.-L.L.; Methodology: K.C.D., S.-L.L.; Project administration: K.C.D., S.-L.L.; Resources: K.C.D., S.-L.L.; Software: K.C.D., S.-L.L.; Supervision: S.-L.L., C.Q.D.; Validation: K.C.D., S.-L.L.; Visualization: K.C.D., S.-L.L.; Writing – original draft: K.C.D., C.Q.D.; Writing – review & editing: K.C.D., S.-L.L., C.Q.D.

**Funding**
Funding was provided by the Howard Hughes Medical Institute (C.Q.D. and S.-L.L.) and the National Institutes of Health (HD27056 to C.Q.D. and K.C.D.). Open Access funding provided by the Howard Hughes Medical Institute. Deposited in PMC for immediate release.

DEVELOPMENT

## Data and resource availability

All relevant data and details of resources can be found within the article and its supplementary information.

## Peer review history

The peer review history is available online at https://journals.biologists.com/dev/lookup/doi/10.1242/dev.205118.reviewer-comments.pdf

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
