## [Peer Review File · Development (Cambridge, England)]

Fer3 is uniquely expressed in Notch^{OFF} hemilineages, where it promotes interneuron identity

Kasey C. Drake, Sen-Lin Lai and Chris Q. Doe
DOI: 10.1242/dev.205118

Editor: Debra L Silver

Review timeline

Original submission:	23 July 2025
Editorial decision:	16 September 2025
First revision received:	25 September 2025
Accepted:	13 October 2025

Original submission

First decision letter

MS ID#: dev.205118

MS TITLE: Fer3 is uniquely expressed in Notch^{OFF} hemilineages where it promotes interneuron identity

AUTHORS: Kasey C. Drake; Sen-Lin Lai; Chris Q. Doe

Dear Dr Doe,

I have now received all the referees' reports on the above manuscript, and have reached a decision. The referees' comments are appended below, or you can access them online: please go to:

As you will see, the referees express considerable interest in your work, but have some significant criticisms and recommend a substantial revision of your manuscript before we can consider publication. Reviewer 1 notes the need for additional functional experiments to make this a sufficient conceptual advance. Both Reviewers 1 and 2 suggest experiments, not all of which are needed. In general, inclusion of some data to further detail the impact of Fer3 on cell fate and function will be important to address concerns of reviewer 1 and others. If you are able to revise the manuscript along the lines suggested, which may involve further experiments, I will be happy to receive a revised version of the manuscript. Your revised paper will be re-reviewed by one or more of the original referees, and acceptance of your manuscript will depend on your addressing satisfactorily the reviewers' major concerns. Please also note that Development will normally permit only one round of major revision. If it would be helpful, you are welcome to contact us to discuss your revision in greater detail. Please send us a point-by-point response indicating your plans for addressing the referees' comments, and we will look over this and provide further guidance.

Please attend to all of the reviewers' comments and ensure that you upload both a 'clean' version of your Word file, along with a highlighted version clearly showing where you have made changes in the revised manuscript. Please avoid using 'Tracked changes' in Word files as these are lost in PDF conversion. I should be grateful if you would also provide a point-by-point response detailing how you have dealt with the points raised by the reviewers in the 'Response to Reviewers' box. If you do not agree with any of their criticisms or suggestions please explain clearly why this is so.

Reviewer 1

SUMMARY OF THE ADVANCE MADE IN THIS PAPER AND ITS POTENTIAL SIGNIFICANCE TO THE FIELD

The manuscript by Doe and colleagues describes the first gene that is expressed exclusively in Notch (OFF) cells in neuroblast lineages in the *Drosophila* embryo. The authors show that Fer3 is expressed in Notch OFF cells in a the NB7-1, NB6-1 and NB5-2 lineages. They also show that Fer3 is expressed in both early and late born cells. The authors further show that Fer3 is repressed by Notch and is itself upstream of Dbx. The experiments are clear and the results support the proposed conclusions.

SUGGESTIONS TO AUTHORS

The paper does not really make enough of a conceptual advance to justify publication in *Development* at this stage. While the authors have done a nice job of showing that Fer3 is expressed in early and late Notch OFF cells there is little information in the manuscript about what is the consequence of manipulating Fer3 in the various neuroblast lineages.

I think the authors should either expand the scope of the paper or send it to a more specialized journal such as *Developmental Dynamics* or *Genesis*, the *Journal of Genetics and Development*. If the authors decide to expand the paper, then a more detailed analysis of what happens to the interneuron in Fer3 mutants would be important. For instance, in a Fer3 mutant does the Notch OFF interneuron switch its fate and become a Notch ON cell? Generally I am asking what happens to the cell itself? What is its fate? Where does it project its axons? Likewise, one can ask what happens if you forcibly express Fer3 in Notch ON cells? Or what happens if you express Fer3 in lineages that normally do not express Fer3. This is a very mature field with a wealth of available reagents so it should be relatively straightforward to do the experiments that I am suggesting.

Reviewer 2

SUMMARY OF THE ADVANCE MADE IN THIS PAPER AND ITS POTENTIAL SIGNIFICANCE TO THE FIELD

This study addresses neuronal specification in the developing *Drosophila* CNS, with particular focus on the role of the Fer3 transcription factor in the final asymmetric cell divisions in three specific lineages. They find that Fer3 is activated in NotchOFF siblings and is both necessary and sufficient to activate the expression of the Dbx transcription factor. Several findings make this study interesting for many developmental neurobiologists. First, while asymmetric cell division is by now widely found and appreciated when it comes to the asymmetric division of neural progenitor cells, it is much less studied in many systems when it comes to the final division of secondary and tertiary progenitor cells. Second, the finding that asymmetric divisions at multiple stages of lineage development, including the final divisions into two neurons, are all gated by asymmetric Notch signalling is not broadly appreciated. Third, all studies in *Drosophila* to date have focused on the NotchON siblings, due to the absence of markers for the NotchOFF siblings. The data are clear and the interpretations sound. I have only one major comment and some minor comments that could help improve the manuscript.

SUGGESTIONS TO AUTHORS

Major comment

1) What other markers could they use in addition to Dbx. Does Fer3 govern other interneuron properties, such as axon pathfinding and/or neurotransmitter identity? This is the only weak part of the paper. What happens when Fer and/or Fer:AD is misexpressed with an NB-specific or broad Gal4 line, and an axonal marker is co-expressed, are there changes to the axonal projections?

Minor comments

2) Introduction: It is of course good to have the introduction quickly funnel into the topic at hand. Having said that, I still feel that the intro is a bit myopic. First, Notch acts at multiple steps of neurogenesis: NB selection, Type I-to-0 division mode switch and asymmetric division of the final

siblings from a GMC. Second, the *Drosophila* nerve cord and its well-defined repertoire of NBs could perhaps be better introduced.

- 3) Results, rows 69-76 and subsequently: Please describe the segments under study, A5, T3, etc. Also, why were these specific segments chosen, and what does it look like regarding Fer3 expression and function in other parts of the CNS?
- 4) Fig 1B-C": the staining for Fer3 and Elav look very grainy, compared to subsequent images where at least Fer3 looks like a typical TF stain.
- 5) Figure 3 and accompanying Fig legend and Results text: Different terms are used to describe Notch activation (Notch[intra], Notch[ME]) or Notch mutation (Notch[LOF], *sanpodo*). I suggest sticking to one term for each scenario.
- 6) Semantics, but I think that technically, Notch[intra] and Fer3:AD are not "misexpressed", they are dominant-activated engineered transgenes, so they are "expressed".
- 7) Row 158-159: A bit confusing, "total labelled cells" given several markers used. Maybe rephrase to "total number of labelled cells in MN7-1/NB5-2 lineage". It is more clearly described in the next section.
- 8) Figure 4K-N: Fer3:AD triggers the generation of more cells in the NB5-6 lineage. This lineage undergoes the Type I-to-0 daughter proliferation switch, which is triggered by NotchON in the NB. It is possible that Fer3:AD, being to some extent a proneural bHLH, disturbs this switch, and hence the extra cells may be due to a failed Type I-to-0 switch. Of course, the likelihood of this notion being true depends upon the segment under study, since NB5-6 makes 5 direct-neurons in T1-T3, only 1 in A1-A7 and 6 in A8-A10.

Reviewer 3

SUMMARY OF THE ADVANCE MADE IN THIS PAPER AND ITS POTENTIAL SIGNIFICANCE TO THE FIELD

Drake et al. characterizes the expression and function of Fer3 protein, a bHLH transcription factor, in the developing *Drosophila* ventral nerve cord (VNC). The authors clearly show that Fer3 is only expressed in Notch-OFF sibling neurons arising from the Notch-mediated asymmetric cell division in all of the three distinct neural stem cell lineages, the identity of which was documented clearly during this study. Drake et al. next do loss and gain of function analysis and in an elegant fashion they show that Fer3 expression is necessary and sufficient for the expression of *Dbx*, another transcription factor previously shown to be expressed in Notch-OFF sibling neurons of these three lineages. Of note, work here strongly suggests that Fer3 acts as transcriptional activator, not a repressor as shown previously by other groups in other cell types, pointing out the importance of the cellular context. Overall, the manuscript is written well in a concise way and is well-fit for publication in the journal *Development*.

SUGGESTIONS TO AUTHORS

Line 77-93: NB7-1 and NB5-2 driver are mentioned to be specific; how about their coverage? Do they target the expected number of progenies for each NB?

Line 74-76: would be good to show the entire nerve cord from a late-stage embryo to if any segmental differences present.

Line 102-103: other studies characterized NB6-1 lineage showing that embryonic NB6-1 generate *Unc-4+* Notch ON neurons and *Dbx+* Notch-OFF neurons. Considering the close relationship of Fer3 and *Dbx*, it is good to mention the presence of *Dbx+* progeny here with proper citations.

Line 329: HCR might not clear for everyone; mention it is a method of RNA situ hybridization.

Line 457: Seroka et al., 2019 listed twice as a and b. I think they are the same paper.

If the fly line "*Gal4-T2Afd96Ca*" is used in this study first time, it would be good to describe the details here.

First revision

Author response to reviewers' comments

As you will see, the referees express considerable interest in your work, but have some significant criticisms and recommend a substantial revision of your manuscript before we can consider publication. Reviewer 1 notes the need for additional functional experiments to make this a sufficient conceptual advance. Both Reviewers 1 and 2 suggest experiments, not all of which are needed. In general, inclusion of some data to further detail the impact of Fer3 on cell fate and function will be important to address concerns of reviewer 1 and others. If you are able to revise the manuscript along the lines suggested, which may involve further experiments, I will be happy to receive a revised version of the manuscript.

We thank the editor and reviewers for their helpful comments, especially reviewers 2 and 3.

Reviewer 1:

The manuscript by Doe and colleagues describes the first gene that is expressed exclusively in Notch (OFF) cells in neuroblast lineages in the *Drosophila* embryo. The authors show that Fer3 is expressed in Notch OFF cells in the NB7-1, NB6-1 and NB5-2 lineages. They also show that Fer3 is expressed in both early and late born cells. The authors further show that Fer3 is repressed by Notch and is itself upstream of Dbx. The experiments are clear and the results support the proposed conclusions.

Thanks for the positive comments and accurate summary. We particularly appreciate the comment "The experiments are clear and the results support the proposed conclusions".

SUGGESTIONS TO AUTHORS

The paper does not really make enough of a conceptual advance to justify publication in *Development* at this stage. While the authors have done a nice job of showing that Fer3 is expressed in early and late Notch OFF cells there is little information in the manuscript about what is the consequence of manipulating Fer3 in the various neuroblast lineages. I think the authors should either expand the scope of the paper or send it to a more specialized journal such as *Developmental Dynamics* or *Genesis*, the *Journal of Genetics and Development*. If the authors decide to expand the paper, then a more detailed analysis of what happens to the interneuron in Fer3 mutants would be important. For instance, in a Fer3 mutant does the Notch OFF interneuron switch its fate and become a Notch ON cell? Generally I am asking what happens to the cell itself? What is its fate? Where does it project its axons? Likewise, one can ask what happens if you forcibly express Fer3 in Notch ON cells?

Thank you for the comment. In this report, we conduct three manipulation experiments: Figure 3 is the manipulation of Notch, and Figure 4 shows both the loss of function and misexpression of Fer3. Currently, Dbx is the only molecular marker we have for NotchOFF neurons, and thus why we used it as a read-out. Since the Fer3 mutants lack Fer3 and Dbx, we have no way of identifying them. We did test whether loss of Fer3 transformed the NotchOFF interneurons into NotchON sibling Eve+ U motor neurons, but we saw no additional Eve+ motor neurons. We have added a line to clarify this finding in the text (line154-156). This is consistent with our experiments showing Fer3 is downstream of Notch signaling.

Regarding axon projections, we currently do not have a method of labeling just the Fer3+ neuronal projections. We did assay projections of the entire lineage, but did not see significant differences in the overall projection pattern. This is not surprising, because the number of Fer3 neurons in each lineage is low, it may have been obscured by the other 20+ neurons in the lineage. Clearly, more specific labeling tools are needed to make specific assertions about morphology changes.

Or what happens if you express Fer3 in lineages that normally do not express Fer3. This is a very mature field with a wealth of available reagents so it should be relatively straightforward to do the experiments that I am suggesting.

We have done this experiment, misexpressing Fer3 in NB5-6 where Fer3 is normally not expressed.

We show that the number of Dbx+ neurons in this lineage goes from 0 in wild type to ~16 following misexpression of Fer3:AD (see Figure 4K-N). Thus, Fer3 can promote Dbx expression even in lineages that normally lack Fer3 (line 193-203).

Reviewer 2:

This study addresses neuronal specification in the developing *Drosophila* CNS, with particular focus on the role of the Fer3 transcription factor in the final asymmetric cell divisions in three specific lineages. They find that Fer3 is activated in NotchOFF siblings and is both necessary and sufficient to activate the expression of the Dbx transcription factor. Several findings make this study interesting for many developmental neurobiologists. First, while asymmetric cell division is by now widely found and appreciated when it comes to the asymmetric division of neural progenitor cells, it is much less studied in many systems when it comes to the final division of secondary and tertiary progenitor cells. Second, the finding that asymmetric divisions at multiple stages of lineage development, including the final divisions into two neurons, are all gated by asymmetric Notch signaling is not broadly appreciated. Third, all studies in *Drosophila* to date have focused on the NotchON siblings, due to the absence of markers for the NotchOFF siblings. The data are clear and the interpretations sound. I have only one major comment and some minor comments that could help improve the manuscript.

Thanks for the positive comments and accurate summary. We appreciate the comment “Several findings make this study interesting for many developmental neurobiologists”

SUGGESTIONS TO AUTHORS

Major comment

1) What other markers could they use in addition to Dbx. Does Fer3 govern other interneuron properties, such as axon pathfinding and/or neurotransmitter identity? This is the only weak part of the paper. What happens when Fer and/or Fer:AD is misexpressed with an NB-specific or broad Gal4 line, and an axonal marker is co-expressed, are there changes to the axonal projections? *Regarding axon projections, we currently do not have a method of labeling just the Fer3+ neuron axons. Since we are not able to label the single Fer3 neurons, we opted to do a more targeted Fer3 misexpression in single lineages as it is likely to be more interpretable. We did assay the entire lineage projections in loss of function and misexpression experiments but did not see significant differences in the overall projection pattern. Since the number of Fer3 neurons in each lineage is low, more specific labeling tools are needed to make specific assertions about morphology changes.*

Minor comments

2) Introduction: It is of course good to have the introduction quickly funnel into the topic at hand. Having said that, I still feel that the intro is a bit myopic. First, Notch acts at multiple steps of neurogenesis: NB selection, Type I-to-0 division mode switch and asymmetric division of the final siblings from a GMC. Second, the *Drosophila* nerve cord and its well-defined repertoire of NBs could perhaps be better introduced.

We agree and have made the requested change in response to the first comment (“Here we focus on the third step: NotchON/NotchOFF GMC asymmetric division; earlier roles for Notch in NB segregation and type 1>0 NB divisions are addressed elsewhere (Skeath and Thor, 2003.)”) To address the second comment, we have added text and references to illustrate the number of NBs per abdominal hemisegment “...generating 30 molecularly distinct NBs per abdominal hemisegment (Broadus et al., 1995)” - thanks for the excellent comments. We can’t go into more detail and still come in under the 3000-word limit.

3) Results, rows 69-76 and subsequently: Please describe the segments under study, A5, T3, etc. Also, why were these specific segments chosen, and what does it look like regarding Fer3 expression and function in other parts of the CNS?

Good point! The number of hemisegments analyzed is shown in the quantifications. Except for experiments assaying NB5-6 (which was restricted to T1-T3 segments) we performed our analysis on A1-A6 segments. There was no distinguishable difference between the abdominal and thoracic segments and the pattern persisted through L1 (we did not look beyond L1). We added a line (line 78-80) addressing this.

4) Fig 1B-C": the staining for Fer3 and Elav look very grainy, compared to subsequent images where at least Fer3 looks like a typical TF stain.

This image is grainy because it is zoomed in and Elav stains are often grainy. We believe the result is still very clear.

5) Figure 3 and accompanying Fig legend and Results text: Different terms are used to describe Notch activation (Notch[intra], Notch[ME]) or Notch mutation (Notch[LOF], sanpodo). I suggest sticking to one term for each scenario.

We agree, and we now use the same terms consistently throughout.

6) Semantics, but I think that technically, Notch[intra] and Fer3:AD are not "misexpressed", they are dominant-activated engineered transgenes, so they are "expressed".

We think that we are both right! We use "misexpression" when we express the gene outside of its normal expression domain; "over-expression" would be used when we increase gene expression within its normal expression domain.

7) Row 158-159: A bit confusing, "total labelled cells" given several markers used. Maybe rephrase to "total number of labelled cells in MN7-1/NB5-2 lineage". It is more clearly described in the next section.

Thanks for the good point; we changed that sentence to: "The total number of Dbx+ neurons..."

8) Figure 4K-N: Fer3:AD triggers the generation of more cells in the NB5-6 lineage. This lineage undergoes the Type I-to-0 daughter proliferation switch, which is triggered by NotchON in the NB. It is possible that Fer3:AD, being to some extent a proneural bHLH, disturbs this switch, and hence the extra cells may be due to a failed Type I-to-0 switch. Of course, the likelihood of this notion being true depends upon the segment under study, since NB5-6 makes 5 direct- neurons in T1-T3, only 1 in A1-A7 and 6 in A8-A10.

We are sorry, we can't find the relevant reference. In fact, work from the Thor lab directly addressed the issue and found no crosstalk: "we found no regulatory interactions with Notch signaling in the NB-5-6T, nor with regulators in the asymmetric NB-daughter machinery (Baumgardt et al., 2007, PlosBiology). Furthermore, we assayed only T1-T3 for this experiment, and these segments do not generate NotchON/NotchOFF siblings.

Reviewer 3:

Drake et al. characterizes the expression and function of Fer3 protein, a bHLH transcription factor, in the developing Drosophila ventral nerve cord (VNC). The authors clearly show that Fer3 is only expressed in Notch-OFF sibling neurons arising from the Notch-mediated asymmetric cell division in all of the three distinct neural stem cell lineages, the identity of which was documented clearly during this study. Drake et al. next do loss and gain of function analysis and in an elegant fashion they show that Fer3 expression is necessary and sufficient for the expression of Dbx, another transcription factor previously shown to be expressed in Notch-OFF sibling neurons of these three lineages. Of note, work here strongly suggest that Fer3 acts as transcriptional activator, not a repressor as shown previously by other groups in other cell types, pointing out the important of the cellular context. Overall, the manuscript is written well in a concise way and is well-fit for publication in the journal Development.

Thanks for the positive comments and accurate summary. We especially appreciate the comment "the manuscript is written well in a concise way and is well-fit for publication in the journal Development".

SUGGESTIONS TO AUTHORS

Line 77-93: NB7-1 and NB5-2 driver are mentioned to be specific; how about their coverage? Do they target the expected number of progenies for each NB?

Great question! The NB5-2 driver is expressed in ~26 neurons, and the NB7-1 driver is expressed in ~24 neurons (Figure 4E, J). Similarly, Dil labeled clones show NB5-2 making 17-26 neurons and NB7-1 making 16-22 neurons (Bossing et al., 1996, Dev Biol). Thus, the coverage is good.

Line 74-76: would be good to show the entire nerve cord from a late-stage embryo to if any segmental differences present.

We restricted our analysis to A1-A6 which is now mentioned in the figure legends. There was no difference in Fer3+ cell number or distribution pattern between abdominal and thoracic segments. There were some Fer3+ cells in the central brain (but we did not look into those). Furthermore, the Fer3+ cell number and distribution pattern remained through L1 (did not look past L1).

Line 102-103: other studies characterized NB6-1 lineage showing that embryonic NB6-1 generate Unc-4+ Notch ON neurons and Dbx+ Notch-OFF neurons. Considering the close relationship of Fer3 and Dbx, it is good to mention the presence of Dbx+ progeny here with proper citations.
Thanks for the comment! We have added this information on new lines 103-104.

Line 329: HCR might not clear for everyone; mention it is a method of RNA situ hybridization.
We agree and have added a sentence as requested "Fer3 HCR (Hybridization Chain Reaction; a method for visualizing RNA in cells)".

Line 457: Seroka et al., 2019 listed twice as a and b. I think they are the same paper.
You are right; we have corrected the error.

If the fly line "Gal4-T2Afd96Ca" is used in this study first time, it would be good to describe the details here.

We agree that this transgene needs a better description. It is a CRISPR-generated reporter inserting a Gal4 coding sequence into the Fd4 (Flybase: fd96Ca) gene to generate a Gal4 line specifically expressed in NB7-1 and its progeny. We cite its recent publication in BioRxiv (<https://doi.org/10.1101/2025.09.04.674256>).

Second decision letter

MS ID#: dev.205118R1

MS TITLE: Fer3 is uniquely expressed in NotchOFF hemilineages where it promotes interneuron identity

AUTHORS: Kasey C. Drake; Sen-Lin Lai; Chris Q. Doe
ARTICLE TYPE: Research Report

Dear Dr Doe,

I am happy to tell you that your manuscript has been accepted for publication in Development, pending our standard publication integrity checks.

Reviewer 1

SUMMARY OF THE ADVANCE MADE IN THIS PAPER AND ITS POTENTIAL SIGNIFICANCE TO THE FIELD

The authors have addressed my concerns - I recommend publication in Development.

SUGGESTIONS TO AUTHORS

The authors have addressed my concerns - I recommend publication in Development.

Reviewer 2

Referring to my original points:

1) It would have been interesting to learn about additional cell specifying roles of Fer3 in addition to controlling Dbx.

8) Notch gates the Type 1 > Type 0 daughter proliferations switch in NB5-6t, and impairment of Notch signalling results in more cells being generated in NB5-6t (PMID 22241838). It therefore seems reasonable that Fer3, being a bHLH in the proneural superfamily could interfere with Notch signalling and trigger extra cells in the NB5-6t lineage.

Reviewer 3

SUMMARY OF THE ADVANCE MADE IN THIS PAPER AND ITS POTENTIAL SIGNIFICANCE TO THE FIELD

Authors addressed all my concerns and I strongly suggest accepting the revised manuscript for publication.